# Effect of Carbonyl Iron Particle Types on the Structure and Performance of Magnetorheological Elastomers: A Frequency and Strain Dependent Study

**DOI:** 10.3390/polym14194193

**Published:** 2022-10-06

**Authors:** Ayman M. H. Salem, Abdelrahman Ali, Rahizar Bin Ramli, Asan G. A. Muthalif, Sabariah Julai

**Affiliations:** 1Department of Mechanical Engineering, Faculty of Engineering, University of Malaya, Kuala Lumpur 50603, Malaysia; 2Department of Mechanical and Industrial Engineering, College of Engineering, Qatar University, Doha P.O. Box 2713, Qatar

**Keywords:** magnetorheological elastomers, magnetic particles, frequency-dependence, strain-dependence, MR effect, Payne effect, dynamic properties

## Abstract

Magnetorheological elastomers (MREs) are smart viscoelastic materials in which their physical properties can be altered when subjected to a varying magnetic field strength. MREs consist of an elastomeric matrix mixed with magnetic particles, typically carbonyl iron particles (CIPs). The magnetic field-responsive property of MREs have led to their wide exposure in research. The potential development and commercialization of MRE-based devices requires extensive investigation to identify the essential factors that can affect their properties. For this reason, this research aims to investigate the impact of CIPs’ type, concentration and coating on the rheological and mechanical properties of MREs. Isotropic MREs are fabricated with four different CIP compositions differing between hard or soft, and coated or uncoated samples. Each MRE composition have three different concentrations, which is 5%, 10%, and 20% by volume. The dynamic properties of the fabricated samples are tested by compression oscillations on a dynamic mechanical analyzer (DMA). Frequency and strain dependent measurements are performed to obtain the storage and loss modulus under different excitation frequencies and strain amplitudes. The emphasis is on the magnetorheological (MR) effect and the Payne effect which are an intrinsic characteristics of MREs. The effect of the CIPs’ type, coating, and concentration on the MR and Payne effect of MREs are elucidated. Overall, it is observed that, the storage and loss modulus exhibit a strong dependence on both the frequency excitations and the strain amplitudes. Samples with hard and coated CIPs tend to have a higher MR effect than other samples. A decrease in the storage modulus and non-monotonous behavior of the loss modulus with increasing strain amplitude are observed, indicating the Payne effect. The results of this study can aid in the characterization of MREs and the proper selection of CIPs grades based on the application.

## 1. Introduction

Magnetorheological elastomers (MREs) are a class of smart viscoelastic composites that exhibit tunable dynamic properties when subjected to a regulated external magnetic field. MREs consist of micron-sized ferromagnetic particles dispersed in elastomeric matrix, typically silicone rubber. The rheological and mechanical characteristics such as yield stress, stiffness and damping are among those properties that can be alerted in MREs. When exposed to an external magnetic field, the magnetic particles within MREs are compactly packed. The magnetic dipole interaction between the magnetized particles increases along the direction of the magnetic field lines. Consequently, the properties of MRE, such as stiffness and damping, can be controlled [1]. The aforementioned properties can be reversibly altered when the magnetic field is removed, where the inter-particle attractions of the magnetic particles vanish; hence the original microstructure of MREs is recovered immediately [2]. For this reason, MREs can be successfully adopted in applications where the controllable and reversible variation of the effective stiffness and damping of a device are desired. Such applications include MREs dynamic vibration absorbers (DVAs) and absorbers [3,4,5,6], engine mounts [7,8], sound insulation [9], and smart actuators [10]. The magneto-mechanical characteristics of MREs in the absence and presence of magnetic fields have been explored in literature [11,12]. These studies are essential in identifying the possible practical uses of MREs in engineering applications. Magneto-mechanical characterizations of MREs include static, dynamic, compression, tensile, and shear tests in MREs’ on and off-field states. Furthermore, studies related to MRE also included investigations of the fatigue and deformation properties of MREs [13]. Therefore, it can be deduced that MREs have received a great deal of interest in research and in various engineering applications.

Silicone rubber, polybutadiene rubber, nitrile rubber, polyurethane rubber, and others make up a typical rubber matrix in MREs [14,15,16]. Moreover, several additives were considered in the fabrication of MREs, such as plasticizers, carbon-based additives, and magnetic and non-magnetic fillers. High magnetic permeability particles are added to the elastomeric matrix during the fabrication process of MRE. The magnetic particles added to the matrix material must have high saturation magnetization and high inter-particle connection. Carbonyl iron particles (CIPs) are one of many magnetically permeable components. CIPs were developed by BASF in 1925 and formulated through the thermal decomposition of iron pentacarbonyl $\left(\mathrm{Fe}{\left(\mathrm{CO}\right)}_{5}\right)$ [17]. CIPs are the most widely adopted type of magnetic particle in the development of MREs. This is because CIPs have high magnetic saturation and high magnetic permeability. Generally, the scattering of the magnetic particles inside the elastomer matrix can either be isotropic or anisotropic. The curing process of the isotropic MREs is performed without an externally applied magnetic field. Contrary to isotropic MREs, the anisotropic class is cured under a magnetic field where the particles are pre-aligned within the matrix.

The tunable dynamic viscoelastic properties of MREs are referred to as the magnetorheological (MR) effect. It is a method to characterize the performance of MREs based on their rheological and mechanical properties. The MR effect is used to express the alteration of the rheological behavior of MREs under the influence of an applied magnetic field [18,19]. Studies involving MRE characterization have reported that the MR effect is affected by various factors, including magnetic field strength, magnetic particles size and content, type of the matrix material, and the frequency oscillation [20,21]. The interfacial interaction between the magnetic particles and the elastomeric matrix considerably affects the dynamic viscoelasticity and the MR effect of MREs [22]. For this reason, the interaction between the particles have been extensively investigated in literature to seek enhancements in the MR effect. The inclusion of additives can substantially enhance the field dependent characteristics of MREs. Khairi et al. [23] added silicone oil as a plasticizer additive to the elastomeric matrix of MRE and successfully achievesd an increased particle alignment and MR effect. Li et al. [24] reported that the mechanical properties of MREs can be enhanced by adding carbon nanotubes (CNTs) to the elastomer matrix, which enhanced their MR effect. Other Studies have reported a significant improve of the MR effect in MREs by adding magnetic fillers as additives. Lee et al. [25] investigated the effect of adding nano-sized gamma-ferrite particles as magnetic additives and reported higher dynamic modulus due to more uniform alignment of the CIPs within the elastomeric matrix.

Another performance parameter to characterize MREs is the Payne effect, which is a particular feature of stress-strain of filled rubber such as MREs. The Payne effect appears as a function of the storage and loss modulus on the amplitude of the applied strain. Above specific strain amplitude values, the storage modulus tends to decrease rapidly with the increasing strain, while the loss modulus shows a maximum value when the storage modulus decreases. Physically, the Payne effect is realized by the deformation changes occurring within the MRE microstructure, where the breakage and recovery of the physical bonds link the adjacent filler particles. There have been studies investigating the Payne effect of MREs in recent years, which indicated the strong dependence of storage and loss modulus on the strain amplitudes. Sorokin et al. [26] investigated the impact of matrix elasticity, filler content, and particles alignment on the dynamic behavior of MREs. It was observed that the Payne effect significantly increased when the material was subjected to magnetic field. Tong et al. [27] investigated the effect of flower-like cobalt particles on the Payne effect and damping properties of MREs. The respective MREs have exhibited higher storage modulus and higher energy dissipation, as well as a reduced Payne effect due to higher crosslink density. Yu et al. [28] fabricated MRE containing polyaniline modified CIPs and reported lower magnitudes of Payne effect due to the enhanced interface between the CIPs and the matrix.

In this work, an experimental characterization of isotropic MREs containing different CIPs compositions is conducted. The aim is to investigate the effect of the magnetic filler concentration, type, and coating on the mechanical and rheological properties of MREs. For this reason, isotropic silicone rubber-based MREs mixed with different CIPs are fabricated and tested using a dynamic mechanical analyzer. Four different grades of CIPs are used in this study differing in surface roughness (hard or soft), surface treatment (coating with silica), and concentration. Frequency-dependent and strain-dependent measurements are performed to obtain the storage and loss modulus, which are later analyzed to examine the dependence of the dynamic properties of the isotropic MREs on the frequency, strain amplitude and magnetic field. The experimental results are used to characterize the performance of the fabricated MRE samples and provide a clear demonstration of their MR and Payne effects. Ultimately, the results can be exploited in the fabrication of MREs to provide a base for the proper selection of the CIPs’ grades which can be helpful in the development of MRE-based devices.

The paper is organized as follows. Section 2 presents the experimental methodology and discusses the material used in the MRE fabrication, the experimental setup used to conduct the dynamic testing, and the experimental measurements. Section 3 provides a comprehensive analysis and discussion of the dependencies of storage and loss modulus on the frequency and strain. It also provides analysis and discussions of the performance of the samples in terms of the MR and Payne effect. The summary and concluding remarks are highlighted in Section 4.

## 2. Experimental

### 2.1. Materials

The two main components in the fabrication of MREs are silicone rubber and CIPs. The silicone rubber is referred to as the elastomeric matrix, and the type used in this study is Elite Double 32 Fast from Zhermack (Badia Polesine, Italy). The characteristics of this type of silicone include high fluidity, high dimensional stability over time, and high elastic recovery. Its advantages also include the fast setting time which is from 10–15 min. For this reason, the sedimentation of the magnetic particles while curing is minimized due to the fast-curing time of the silicone. Additionally, the high fluidity of this silicone enables mixing without vacuum conditions which facilitate the fabrication process of the MRE. The silicone elastomeric matrix consists of a silicone base and catalyst mixed with 1:1 mixing ratio. The properties of the silicone rubber used in this study are provided in Table 1.

As for the CIPs, they are formed by thermal decomposition of iron pentacarbonyl FeCO_5_. The CIPs developed by BASF have high magnetic saturation and high magnetic permeability due to the use of ferromagnetic materials such as iron. These characteristics are essential in the development of MRE materials to ensure high levels of magnetization in response to an applied magnetic field. The characteristics of MREs are highly influenced by their magnetic fillers, such as CIPs. The addition of CIPs in the elastomeric matrix can significantly increase the stiffness and hardness of MREs. For this reason, an optimal volume fraction of CIPs must be used to ensure sufficient magnetization of the MREs. Soft magnetic fillers usually have low coercivity and high magnetic permeability, which make them easier to magnetize and demagnetize, while hard fillers can retain their magnetism for long-term after the removal of the applied magnetic field. This is because of the high coercivity and larger magnetic fields required for magnetization to saturation. One main issue accompanied by the inclusion of CIPs in MREs development is the compatibility between the magnetic particles and the matrix material. The matrix materials are hydrophobic, whereas CIPs are generally hydrophilic. For this reason, studies have explored ways to increase the compatibility between magnetic particles (CIPs) and the matrix material (elastomer) [29,30]. One way is the surface treatment of CIPs with silane coupling agents. It is reported that coating the magnetic particles can significantly increase the affinity and compatibility, increase the bond strength with the elastomeric matrix, and reduce the sedimentation of the magnetic particles [31,32].

In this study, different types of carbonyl iron particles are used as magnetic fillers differing between soft and hard magnetic particles, which are either coated or not with silicone dioxide silica (SiO_2_). The CIPs used are polydisperse with varying particle size distribution (Avg. D50: 2–6.8 μm). The fabrication steps of soft and hard CIPs are shown in Figure 1. More details on the properties of the used CIPs are provided in Table 2. Crude iron is reacted with carbon dioxide to produce iron pentacarbonyl (IPC), which is decomposed to rase, hard CIP. Silica coating is added to the particles when needed. The soft CIPs are produced by heating the hard CIP in an oven [33].

MRE samples are fabricated with different combinations of soft, hard, coated magnetic particles and CIPs volume fractions. The fabricated MRE samples are isotropic in which the curing process is performed in the absence of an applied magnetic field where the magnetic particles are distributed homogeneously within the elastomeric matrix. The polarization process can be carried out in the presence of magnetic field. In this case, the magnetic particles are cured and aligned in the direction of the magnetic field, forming anisotropic MREs where the arrangement of the magnetic particles form a chain-like structure within the matrix. The fabrication process of the MREs samples is performed by three main steps: Mixing, molding, and curing without magnetic field application. Firstly, the silicone base is mixed with certain volume fractions of CIPs (5%, 10% and 20%). The mass of the CIPs required for the mixture is calculated using the following formula: (1)$$m_{CIPs}=\rho_{CIPs}\;\left(VF\%\times V_{T}\right)$$
where $\rho_{CIPs}$ is the density of the *CIPs*, VF% is the volume fraction, and $V_{T}$ is the total volume of the mixture measured in mL. Both parts are adequately mixed, and the silicone catalyst is added to the mixture and mixed manually for 1 min. If mixing is carried out mechanically with vacuum mixture the mixing time is reduced to 30 s. Next, the mixture is poured into the casting mold and left to cure for 15 min (working time: 5 min; setting time: 10 min). The excess MRE liquid can flow from the casting mold through the risers located in the mold’s upper plate. The risers prevent cavities from forming inside the MRE due to shrinkage during the curing process. Each sample has a diameter of 25 mm and height of 8 mm. The aforementioned process is repeated for each of the 12 MRE samples. The process is clearly presented in Figure 2, and details about the fabricated specimens are provided in Table 3.

### 2.2. SEM Characterization

Figure 3 shows the SEM images captured for the different grades of carbonyl iron particles used in this study where (a) hard-coated, (b) soft-coated, (d) hard-noncoated, and (c) soft-noncoated particles. The white dots shown in Figure 3a,b are the silica (SiO_2_) coating where the particles in Figure 3a have a rough surface, and the particles in Figure 3b have a smooth surface. These particles have approximately the same particle size. The hard-noncoated particles shown in Figure 3c are 30–50% smaller than the other samples in terms of particle size and have a rough surface. The particles in Figure 3d possess the largest particle size among the other types and have a smooth surface. All the CIP grades mentioned above are mostly spherical with varying particle size distribution. It must be noted that the particles shown in Figure 3 are solely images of the CIPs in powder form. The morphological characteristics of the fabricated MRE samples are presented in Figure 4. The fabricated samples have random particle distribution associated with isotropic MREs. As stated earlier, this is because of the absence of a magnetic field during the curing process. The structure of the elastomeric matrix is shown in Figure 4a, which corresponds to the unfilled MRE sample with a 0% CIP volume fraction. While Figure 4b,c show the MRE samples with 10% and 20% volume fractions, respectively. From observing the SEM images, it can be deduced that the MRE samples possess a homogeneous particle dispersion between the rubber matrix and the CIPs. However, the CIPs have more tendency to clump together at higher volume fractions, as shown in Figure 4c. Other factors, such as particle size and surface area, can contribute to particle clumping. Additionally, air gaps are more likely to form with higher rates at 20% volume fractions due to rubber mixing and processing difficulties.

### 2.3. Measurements

The dynamic compression test is performed using a commercially available dynamic mechanical analyzer (DMA) (from T.A. instruments, model: RSA-G2). A 3D-printed holder is fabricated and attached to the DMA base plate. The holder is used to fix permanent magnets around the MRE specimen since the testing instrument is not supplied with an electromagnet. The magnets are attached around the specimen and are capable of producing magnetic field perpendicular to the parallel plates. The number of permanent magnets fixed around the MRE specimens controls the magnetic field intensity. The experimental setup used in this work is presented in Figure 5. A finite element analysis simulation is performed to investigate the proper selection of magnetic polarity, which direct the maximum flux into the sample, as shown in Figure 6. The magnetic flux density (B) is measured by a Gauss meter and the highest recorded value at full magnets capacity is 230–340 mT. The disc-shaped samples (diameter: 25 mm; height: 8 mm) are fixed between a lower plate connected to a motor subject to force linear oscillations and a stationary upper plate connected to the transducer. A frequency sweep test is performed where the temperature and strain are held constant, and the frequency is varied. The strain amplitude is fixed at γ = 1% and the frequency (f) was varied in a range of 0.1–25 Hz. The frequency dependencies of the storage modulus ($E^{\prime }$) and loss modulus ($E^{\prime \prime }$) measured in the absence (off-state) and presence (on-state) of magnetic field are recorded and plotted against the frequency. The frequency sweep test is performed to identify the MR effect of the MRE samples. Similarly, a strain sweep test is performed for amplitude oscillations where the frequency and temperature are held constant, and the strain is varied. The frequency is fixed at 10 Hz, while the strain amplitude is varied for a range of 0.05–5%. The strain sweep test is performed to identify the Payne effect of the MRE samples. The specimens are secured properly between the parallel plates, and the strain amplitude was limited to 5% to avoid any slippage between the measuring plates and the sample surfaces. During the measurements, the displacement $u\left(t\right)$ and the force response $F\left(t\right)$ are obtained and are approximately sinusoidal harmonic functions. Hence, the compression stress $\tau \left(t\right)$ and the compression strain $\gamma \left(t\right)$ also vary sinusoidally with their respective amplitudes $\tau_{a}$ and $\gamma_{a}$. However, the stress and strain will lag by a phase angle $\left(\delta \right)$ which is called the loss angle. The dynamic complex modulus ($E^{*})$ is the ratio of the amplitudes. The loss tangent ($\tan\left(\delta \right)$) is the ratio of the loss modulus and storage modulus. The storage modulus indicates the MRE stiffness, as it indicates the ability of MRE to store the deformation energy. The loss modulus represents the ability to dissipate the deformation energy. The method to determine the storage, and loss modulus from the amplitudes and the loss angle is described as follows [34]:(2)$$E^{*}=\frac{\tau_{a}\left(\omega \right)\;}{\gamma_{a}\left(\omega \right)},\;\;\;\;E^{\prime }=E^{*}\cos\left(\delta \right),\;\;\;\;E^{\prime \prime }=E^{*}\sin\left(\delta \right)$$

## 3. Results and Discussion

### 3.1. Dependence of MRE Dynamic Properties on the Frequency

The dynamic property dependence on the frequency of the MRE samples is investigated through a frequency sweep test. In this test, the strain amplitude is held constant at γ = 1%, while the frequency is varied by sweep from 0.1–25 Hz. Frequency dependencies of the storage and loss modulus are shown in Figure 7, measured at the on-state (presented by solid lines) and off-state (presented by dotted lines). The measurements are observed under the limit of the linear viscoelastic region (LVE), representing the range in which the test can be carried out without destroying the structure of the MRE samples. Figure 7a,b are for the samples having volume fraction of 5%, while Figure 7c,d are for 10% volume fractions and Figure 7e,f are for 20% volume fractions. The storage and loss modulus for the unfilled sample are presented in Figure 7a,b. Analyzing the results obtained from the frequency sweep test, the following observation can be made: 

(i)The storage modulus increases with the frequency for all measured samples, which indicates the dependencies of the MRE dynamic properties on the frequency. It is observed that $E^{\prime }$ increased almost linearly with the frequency. Similar observations were reported in previous studies [35,36]. It was also observed that increasing the concentration of the CIPs can significantly increase $E^{\prime }.$ Furthermore, the increase of $E^{\prime }$ is highly influenced by the addition of the CIPs. It is observed from comparing the same samples but with varying CIPs concentrations that, the maximum increase in $E^{\prime }$ can be obtained at higher CIPs percentages. The lowest values of $E^{\prime }$ are observed in the unfilled sample. The samples with hard CIPs contributed to higher $E^{\prime }$ as compared to those with soft CIPs. However, it can be seen that the S-NC samples have the highest $E^{\prime }$, which is due its large magnetic particle size of this CIP grade.(ii)Similarly, it is observed that the $E^{\prime \prime }$ increases with the frequency for all samples. It is also observed that $E^{\prime \prime }$ increases with the increase of the CIP concentrations. However, filling the elastomer matrix with magnetic particles leads to some decrease in the growth rate of $E^{\prime \prime }$. The maximum percentage increase in $E^{\prime \prime }$ is more pronounced in the samples containing soft CIPs such as S-C and S-NC. The silica coating of the CIPs did not show specific variation in the loss modulus of the MRE samples. Additionally, the loss modulus plots for all measured samples deviate from linearity at lower frequencies, similar to the reported results in previous studies [34,37,38].(iii)Both $E^{\prime }$ and $E^{\prime \prime }$ have increased with respect to the external magnetic field in the on-state. However, a decrease in the growth rate of the loss modulus in the on-state is observed. It can be observed that the highest magnetic response is realized at the low frequencies. Several changes in the moduli with respect to the strength of the applied magnetic field have been reported in previous studies [39,40,41,42]. The increase in the magnetic field strength results in an increase in the interaction between the magnetic particles as they form the chains parallel to the field lines. The increased magnetic interaction between the particles leads to more entanglement of the particles which results in higher $E^{\prime }$. It is reported that anisotropic MREs generally exhibit higher dynamic characteristics because of the pre-ordering of some of the magnetic particles during the polarization process. Consequently, the chains become stronger in the presence of the magnetic field, thus increasing the storage modulus. As for the increase in the loss modulus, the increased magnetic interactions in the on-state leads to an increase in the interfacial friction between the magnetic particles and the matrix, which enhance the energy dissipation, thus resulting in higher loss modulus. 

**Figure 7 polymers-14-04193-f007:**
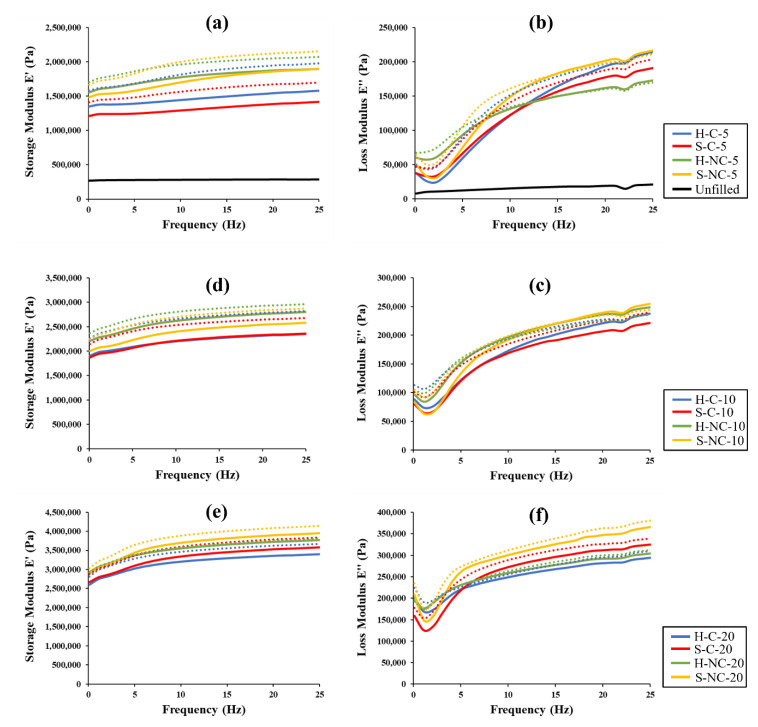
Frequency dependence of storage E′ and loss E″ modulus measured in the off-state (solid lines) and on-state (dotted lines); (**a**) and (**b**): at 5%; (**c**) and (**d**) at 10%; (**e**) and (**f**) at 20% volume fractions.

### 3.2. Magnetorheological Effect

The alteration of MRE dynamic properties in the presence of magnetic field is referred to as the magnetorheological (MR) effect. This effect is an important characteristic of MREs and is used to evaluate their performance. The MR effect is related to the tendency of the CIPs to change their position under the influence of an applied magnetic field. The movement of the magnetic particles due to the application of magnetic field cause deformation in the elastomeric matrix, which alters the dynamic properties of MREs, such as an increase of the stiffness or damping. The deformation of the elastomeric matrix is realized by the interaction of the magnetic particles in the presence of the magnetic field, which brings the magnetic particles closer. The MR effect is described by the absolute and the relative effect. The absolute MR effect ($\Delta E^{\prime }{}_{f})$ is the difference between the maximum storage modulus ($E_{max}^{\prime }$) achieved in the on-state and the storage modulus obtained at the off-state ($E_{0}^{\prime }{}_{f})$. The term $E_{0}^{\prime }$ is referred to as the zero-field modulus, while the ($E_{max,}^{\prime }{}_{f}-E_{0}^{\prime }{}_{f})$ is called the magneto-induced modulus. Hence, the relative MR effect can be obtained by the following expression: (3)$$\mathrm{Relative}\;\mathrm{MR}\;\mathrm{effect}\;\left(\%\right)=\frac{E_{max,}^{\prime }{}_{f}-E_{0}^{\prime }{}_{f}}{E_{0}^{\prime }{}_{f}}\times 100$$

The relative MR effect is dependent on both the frequency and the intensity of the external magnetic field. The frequency dependence of the MR effect is presented in Figure 8. The relative MR effect is calculated from the measurements of the frequency sweep test and is presented as a function of the frequency and the magnetic flux density. The effects of the CIPs’ types, coating and concentration on the MR effect are investigated. It must be noted that the samples are also different in the particle size distribution. Previous studies have reported that the filler size and distribution within the elastomeric matrix influence the MR effect [43,44]. The MR effect was higher in MREs with mixed-size particles, in which magnetic dipole interaction with adjacent particles is increased. This effect is realized for all the MRE samples fabricated in this study because they have polydisperse magnetic particles possessing a wide particle size distribution. Analyzing the results obtained from MR effect, the following observation can be made:(i)The results show that the relative MR effect of all measured samples increases with the frequency and decreases slightly at higher frequencies. At frequencies above 15 Hz, the relative MR effect remains unchanged. A similar observation in a previous study was reported by Nam et al. [34]. It is also observed that as the content of the CIPs increases, the MR effect starts to decrease at even lower frequencies and settles for a larger frequency band.(ii)The MR effect is highly influenced by the type and the coating of the magnetic particles. Hard magnetic particles tend to have a higher MR effect than soft particles. The MR effect is found to be higher in H-C-5, H-C-10 and H-C-20. It is stated that the rough surface of the hard magnetic particles can improve their dispersion stability within the matrix, and hence, enhance their MR properties [45]. The hard magnetic particles also provide MREs with stiffening responses that can be sustained for longer periods even without applying the magnetic field. For this reason, the magneto-induced modulus becomes higher, thus increasing the MR effect.(iii)The results show that coating the magnetic particles with silica can significantly enhance the performance of the MREs. For example, the S-C samples have a higher relative MR effect than the S-NC samples at all concentrations. Similarly, the hard samples have demonstrated a higher MR effect when the particles are coated. However, it can be seen that the effect of CIP types is more pronounced than the effect of the particle coatings. For this reason, the H-C samples have a higher relative MR effect than S-C samples.(iv)The relative MR effect increases with the concentration of the CIPs, however, reaches the magnetic saturation faster at higher concentrations. For this reason, at 20% volume fraction the MR effect was found to be lower as compared to the other samples. Hence, it can be concluded that the CIP concentration is not directly proportional with the MR effect and there exists a specific value at which the MR effect is optimum. From the observations, the optimum volume fraction is found to be at 5%. This can be attributed to the stronger dipole-dipole interaction in the chain between the magnetic particles, which tends to become weaker at higher volume fractions due to the decreased gaps between the particles.

**Figure 8 polymers-14-04193-f008:**
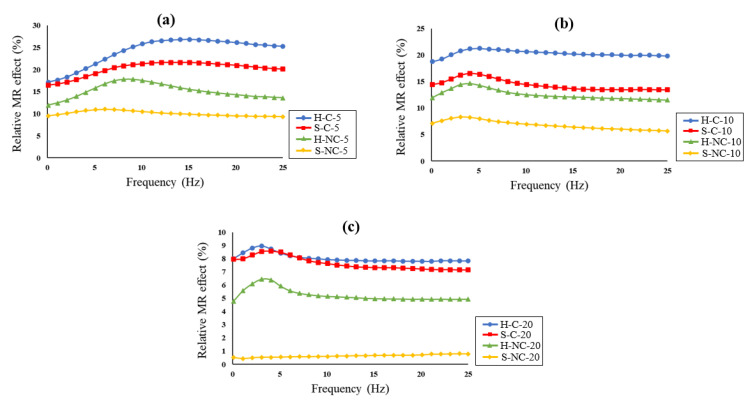
Relative MR effect of the MRE samples: (**a**) at 5%; (**b**) at 10%; (**c**) at 20% volume fractions.

### 3.3. Dependence of MRE Dynamic Properties on the Strain

Following the frequency sweep test, strain amplitude tests are performed in both the on-state and off-state to investigate the dependence of MREs’ dynamic properties on the strain. In this test, the strain amplitude is changed from 0.05% to 5% while frequency is held constant at the intermediate frequency of 10 Hz. The storage and loss modulus are measured and plotted in Figure 9. The measurements recorded in the off-state are presented by the solid lines, while those measured in the on-state are presented by the dotted lines. Figure 9a,b are for the samples having volume fraction of 5%, while Figure 9c,d are for 10% volume fractions and Figure 9e,f are for 20% volume fractions. The measurements show that both moduli of the unfilled sample are independent on the strain amplitude. Some fluctuations in the measurements occur due to a possible slippage resulting from improper placement of the specimen. As for the filled samples, it is observed that both the storage and loss moduli become strain-dependent when the samples are filled with CIPs. It is observed that the storage modulus decreases with the increasing strain amplitude in both the on-state and off-state. For samples containing 5% volume fraction, the linear viscoelastic region proceeds up to γ~1%. As the volume fraction percentage increase to 10%, the non-linear viscoelastic region starts earlier at γ~0.5%. A further increase in the volume fraction to 20% leads to a decrease in the viscoelastic region until γ~0.3%. After the viscoelastic region, the decrease in the storage modulus becomes more pronounced. The responses of the storage modulus start to merge at γ~2.5% for almost all measured samples. It should be noted that this monotonous characteristic behavior of strain dependent $E^{\prime }\left(\gamma \right)$ is similar in all measurements.

The characteristic response of the strain-dependent loss modulus $E^{\prime \prime }\left(\gamma \right)$ is slightly different. It is observed $E^{\prime \prime }$ tends to increase with strain amplitude at the low strain region, and the maximum value of $E^{\prime \prime }\left(\gamma \right)$ appears at 0.4–0.5% strain. This means that an increase in the strain amplitude up to 0.4–0.5% leads to an increase in the loss modulus, while at higher amplitudes from 1–5% the loss modulus starts to decrease. The non-monotonous behavior of the loss modulus with the increasing strain amplitudes is similar for all measured samples. 

The increase in the loss modulus at small γ% levels is realized by the energy required for the continual tearing of the magnetic coupling between the magnetic fillers as the strain amplitude increases. While the subsequent decrease in the loss modulus at the high strain amplitudes can be explained by magnetic structure which breaks at high γ% levels. At higher strain amplitudes, the interactions between the magnetic particles start to decrease because of the increasing distance between the adjacent particles. The non-monotonous behavior of $E^{\prime \prime }\left(\gamma \right)$ of the measured samples H-C, S-C, H-NC, and S-NC is somewhat similar. This is because the elastomeric matrix used to fabricate the samples is the same, which provides similar mobility of the magnetic particle within the matrix, thus almost similar structuring processes. In hard elastomeric matrices, the increase in the loss modulus would be expected to take place throughout a larger strain amplitude region, while would drop significantly at smaller strain amplitude regions in softer matrix types. 

The reduction in $E^{\prime }$ with the increasing strain amplitude is a strong indication to the Payne effect phenomenon which is an attribute of the filled rubber such as MREs. The Payne effect is caused by the destruction-reforming of the magnetic filler network and the bonds linking the silicone rubber network, leading to softening of the MRE due to changes in the microstructure. In the subsequent subsection, the Payne effect for the isotropic MREs with CIPs differing in the type, coating and concentration is discussed in detail.

### 3.4. Payne Effect

The Payne effect phenomenon is related to the dependencies of the storage and loss modulus on the strain amplitude. The appearance of the Payne effect in MRE structures is caused by the breakage and recovery of the bonds formulated between the elastomeric matrix and the magnetic particles. This behavior is also evident when the MREs are subjected to an external magnetic field. The CIPs become more active and are rearranged due to the increase in the magnetic dipole interactions between the adjacent particles when are subjected to the magnetic field. As a result, both $E^{\prime }$ and $E^{\prime \prime }$ increase with the increasing magnetic field intensity, in addition to causing a stronger strain dependence of the moduli. The magnetic particle interactions are manifested in the tendency of the particles to move from the initial equilibrium position to the formation of structures aligned along the magnetic field lines. This interaction between the particles is responsible for strengthening the MRE structure in the presence of the magnetic field. During small deformation (low γ%), the structure formed during the aforementioned interaction is only slightly disrupted, while the large deformations (high γ%), result in completely destroying the filler network and hence the MRE structure becomes softer. Thus, the Payne effect is caused by the gradual breakdown of the magnetic filler within the elastomeric matrix. The Payne effect is measured from the strain-dependent measurements using the initial and final values of the storage modulus and can be calculated using the following expression [31]: (4)$$\mathrm{Payne}\;\mathrm{effect}\;\mathrm{factor}\;\left(\%\right)=\frac{E_{0.05\%}^{\prime }-E_{5\%}^{\prime }}{E_{0.05\%}^{\prime }}\times 100$$

The term $E_{0.05\%}^{\prime }$ is the initial storage modulus, and $E_{5\%}^{\prime }$ is the final storage modulus at the maximum strain amplitude. The term $\left(E_{0.05\%}^{\prime }-E_{5\%}^{\prime }\right)$ can be written as ($\Delta E^{\prime }{}_{\gamma }$) which is the difference between the initial and the final storage modulus. The values of the Payne effect factor for each sample are presented in Table 4 and the results are plotted in Figure 10. The Payne effect factor is increases with respect to the strength of the applied magnetic field. This is because the increase in $E^{\prime }$ with the magnetic filed leads to an increase in $\Delta E^{\prime }{}_{\gamma }$, thus the Payne effect factor becomes greater. The changes in the Payne effect results from the deformations occurring within the MRE structure during strain. Many parameters can influence the Payne effect in MREs, such as the filler size and distribution, the matrix elasticity, and the crosslink density [27]. This section analyzes the influence of CIP types, coating and concentration on the Payne effect in detail. Following the analysis, the observations below can be made:(i)It is observed that, the induced Payne effect is enhanced with the increasing concentration of the CIPs. For instance, the Payne effect for H-C-20 sample is higher than that of H-C-5 and H-C-10. This can be attributed to the increased interactions at higher volume fractions. It is also observed that the silica coating has an influence on the Payne effect. At the same concentrations, the samples with silica coating have lower Payne effect than the noncoated samples. This is because of the increased affinity between the CIPs and the matrix, which lowers the rates of stiffness decrease at higher strain deformations.(ii)Subjecting the MRE samples to an external magnetic field leads to an increase in the Payne effect. Enhancements of the Payne effect due to the application of the magnetic field are observed for every sample. This can be attributed to the increase in the storage modulus due to the increased interaction between the magnetic particles as they form the chains parallel to the field lines. Additionally, it is observed that samples with soft CIPs (such as S-NC-5) have higher Payne effect than those with hard CIPs (such as H-NC-5). This is because the MRE samples with hard CIPs are difficult to deform due to the sustained stiffening behavior. A summary of the results and analysis is provided in Table 5.

**Table 4 polymers-14-04193-t004:** The initial and final storage modulus and the Payne effect factor for the measured samples.

Sample	Payne Effect Factor					
	$E_{0.05\%}^{\prime }$		$E_{5\%}^{\prime }$		Payne Effect (%)	
	Off-State	On-State	Off-State	On-State	Off-State	On-State
H-C-5	2.306	2.531	0.831	0.874	64.0%	65.5%
S-C-5	2.177	2.399	0.833	0.870	61.7%	63.7%
H-NC-5	2.278	2.436	0.834	0.869	63.4%	64.3%
S-NC-5	2.482	2.685	0.842	0.895	66.1%	66.7%
H-C-10	2.610	2.909	0.870	0.920	66.6%	68.4%
S-C-10	2.593	2.890	0.866	0.907	66.6%	68.6%
H-NC-10	2.866	3.032	0.871	0.897	69.6%	70.4%
S-NC-10	2.937	3.221	0.885	0.933	69.9%	71.0%
H-C-20	3.270	3.621	0.922	0.933	71.8%	74.2%
S-C-20	3.631	3.953	0.931	0.944	74.4%	76.1%
H-NC-20	3.378	3.765	0.920	0.955	72.8%	74.6%
S-NC-20	3.960	4.402	0.929	0.966	76.5%	78.0%

**Table 5 polymers-14-04193-t005:** Results and analysis summary.

	Frequency-Dependence	Strain-Dependence	MR Effect	Payne Effect
H-C	Higher $E^{\prime }$ than soft CIPs	Monotonous behavior of $E^{\prime }$ and non-monotonous for $E^{\prime \prime }$	Higher MR effect than soft CIPs. Higher MR effect than H-NC. The effect of CIP type is more pronounced than the effect of the coating.	Lower Payne effect compared to H-NC at same VF%
S-C	Higher $E^{\prime \prime }$ than hard CIPs	Monotonous behavior of $E^{\prime }$ and non-monotonous for $E^{\prime \prime }$	Higher MR effect than S-NC	Lower Payne effect compared to S-NC at same VF%
H-NC	Higher $E^{\prime }$ than soft CIPs	Monotonous behavior of $E^{\prime }$ and non-monotonous for $E^{\prime \prime }$	Higher MR effect than soft	Lowest Payne effect at the same VF%
S-NC	Higher $E^{\prime \prime }$ than hard CIPs	Monotonous behavior of $E^{\prime }$ and non-monotonous for $E^{\prime \prime }$	Lowest MR effect	Higher Payne than hard CIPs.
VF%	Higher percentages lead to higher $E^{\prime }$ and $E^{\prime \prime }$	Earlier start of the non-linear viscoelastic region at higher VF%	Optimum value at 5%	Higher percentages lead to a higher Payne effect for each sample
Magnetic field	Both $E^{\prime }$ and $E^{\prime \prime }$ increase with the magnetic field.	Higher $E^{\prime }$ and $E^{\prime \prime }$ at higher intensities.	Higher intensity leads to higher MR effect	Higher intensity leads to higher Payne effect

**Figure 10 polymers-14-04193-f010:**
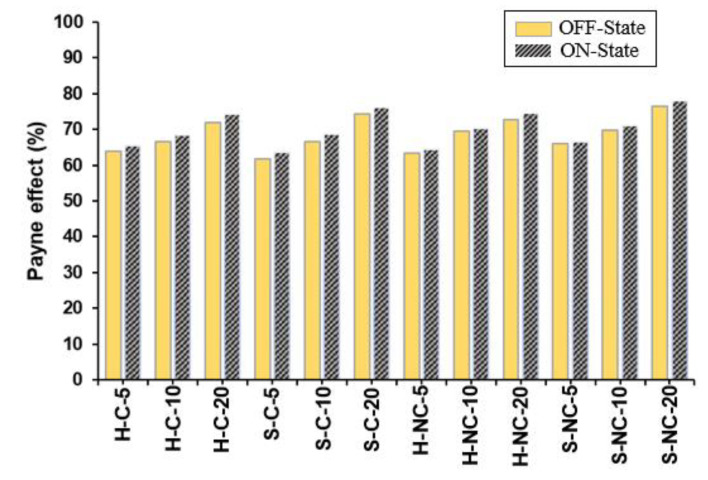
Comparison of the Payne effect of MREs with different combinations of CIP types and volume fractions.

## 4. Conclusions

The experimental characterization of the dynamic properties of isotropic silicone rubber-based MRE with different CIP compositions is conducted in this paper. Frequency and strain dependent measurements are performed on dynamic mechanical analyzer with the samples mounted on parallel-plate compression fixture and subjected to squeeze deformations. The storage and loss modulus of the samples are recorded at various frequencies and strain amplitudes in the absence and presence of an applied external magnetic field. The frequency sweep measurements are performed to examine the frequency dependence of the samples and to highlight their MR effect, while strain amplitude measurements are performed to investigate the strain dependence and to evaluate the Payne effect. The microstructure of the fabricated MRE samples is inspected by SEM characterization analysis, where it is shown that the CIPs have a homogeneous dispersion inside the matrix.

Firstly, the frequency measurements showed a strong dependence of the moduli on the frequency, as the storage and loss modulus increased with the frequency. The storage and loss modulus become higher in MRE samples with hard CIPs and concentrations, hard CIP types. The application of the external magnetic field generated an additional rise in both the storage and loss moduli due to the increased magnetic interactions. Secondly, the MR effect is found to be influenced by the type, concentration and coating of the magnetic particles. It is observed that the hard magnetic particles tend to have a higher MR effect than soft particles. Additionally, coating the magnetic particles with silica is found to enhance the MR effect. Furthermore, it is observed that an increase in the CIPs concentration do not necessarily cause an enhancement in the MR effect. The optimum MR effect is found at CIPs concentration of 5%.

Secondly, the strain measurements showed a reduction in the storage modulus and a non-monotonous behavior of the loss modulus with the increasing strain amplitude. The changes in both the storage and loss moduli with respect to the strain deformations provide a great indication to the Payne effect. The Payne effect is enhanced with the increasing concentration of CIPs. Lower Payne effect values are observed for samples with coated CIPs because the coating enhances the affinity between the MRE particles. Moreover, the MRE samples with hard CIPs showed lower Payne effect due to the better stiffening response of the hard CIPs that can be sustained for longer periods.

Lastly, the results of this study can pave the way for the proper selection of the CIPs in the fabrication of MREs. The performance of the MRE samples with different CIP compositions is highlighted by their respective MR and Payne effects and serve as a benchmark while selecting such materials for the development of MRE-based devices, especially tunable vibration isolators operating on the squeeze mode.

## Figures and Tables

**Figure 1 polymers-14-04193-f001:**
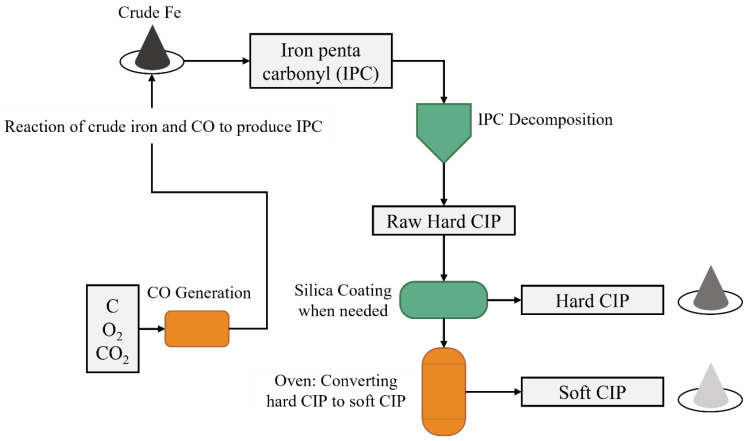
Production technique of soft and hard carbonyl iron particles.

**Figure 2 polymers-14-04193-f002:**
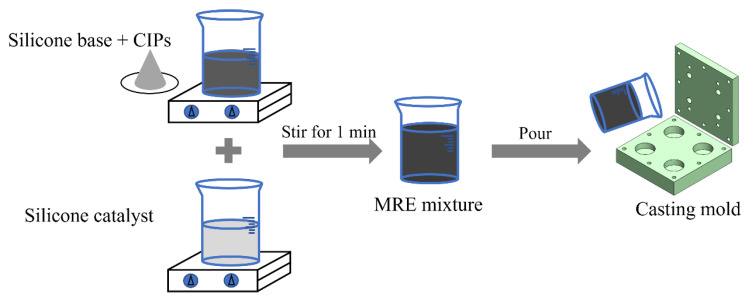
Schematic diagram of MRE fabrication process.

**Figure 3 polymers-14-04193-f003:**
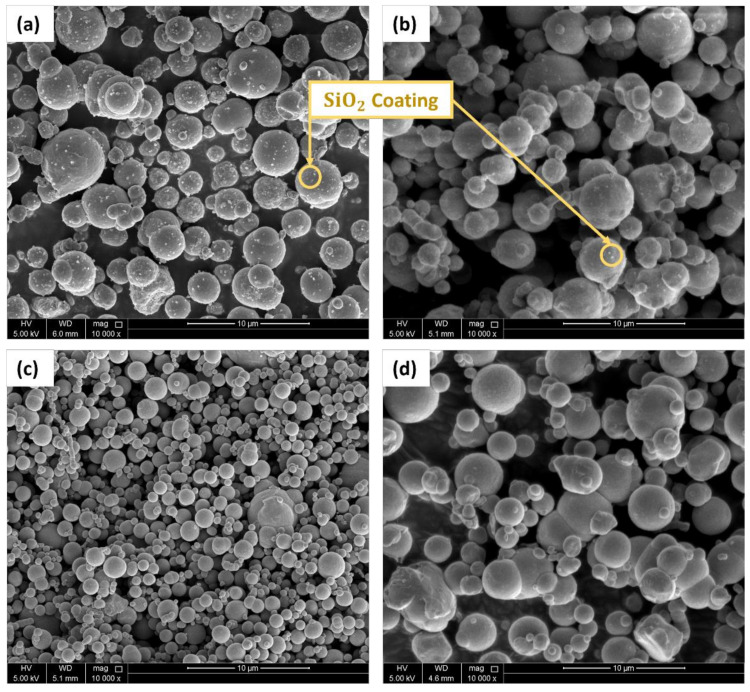
SEM images of the carbonyl iron particles; (**a**) type 1: H-C, (**b**) type 2: S-C, (**c**) type 3: H-NC, and (**d**) type 4: S-NC.

**Figure 4 polymers-14-04193-f004:**
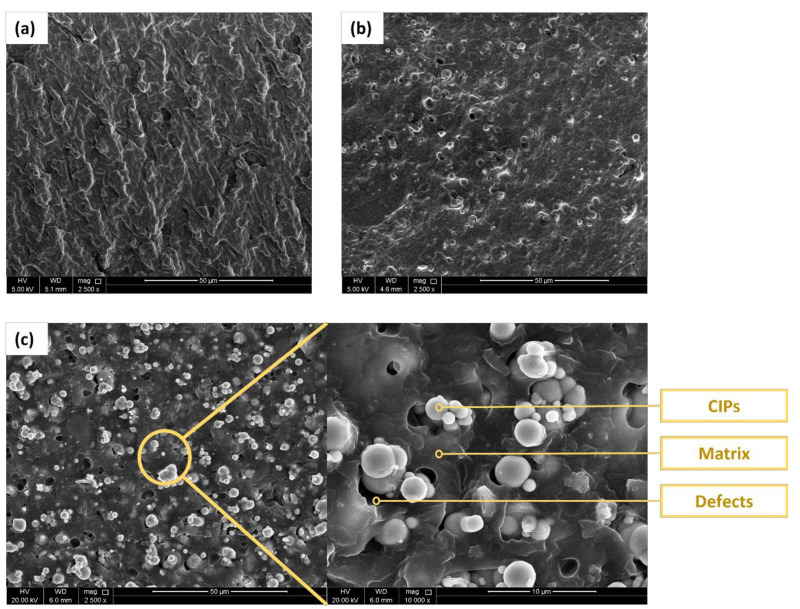
SEM images of isotropic MRE samples; (**a**) unfilled sample, (**b**) with 10% CIP volume fraction, and (**c**) with 20% CIP volume fraction.

**Figure 5 polymers-14-04193-f005:**
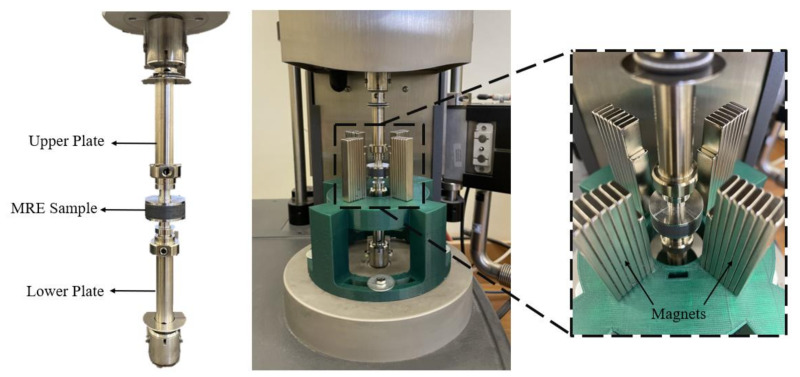
Dynamic Compression test of MRE samples on the dynamic mechanical analyzer.

**Figure 6 polymers-14-04193-f006:**
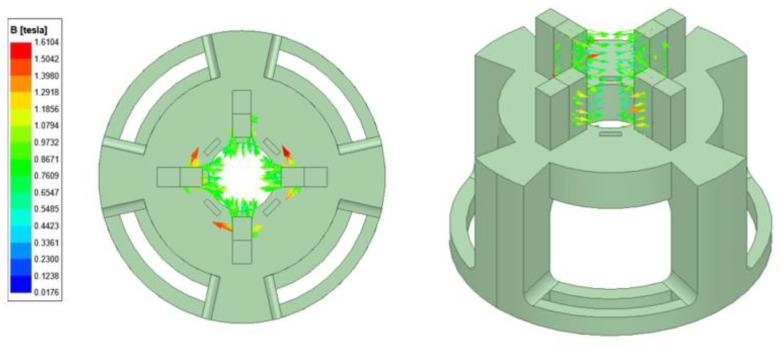
Magnetic vector contour for the magnetic field system by finite element analysis.

**Figure 9 polymers-14-04193-f009:**
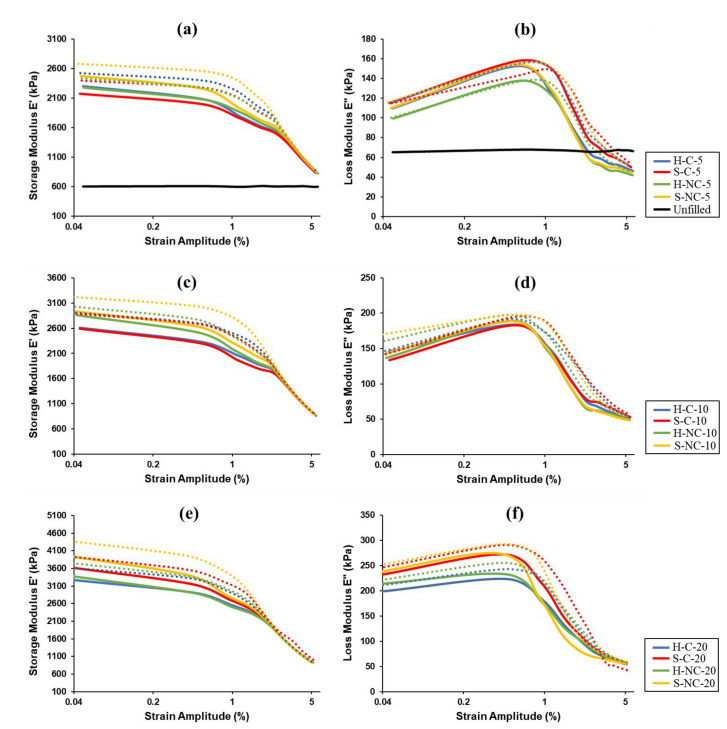
Strain dependence of storage E′ and loss E″ modulus measured in the off-state (solid lines) and on-state (dotted lines); (**a**,**b**): at 5%; (**c**,**d**) at 10%; (**e**,**f**) at 20% volume fractions.

**Table 1 polymers-14-04193-t001:** Material Properties of the silicone rubber.

Material	Properties	
Silicone rubber	Type	Elite Double 32 Fast—Zhermach
	Mixing ratio	1:1
	Manual mixing time (min:sec)	1:00
	Working time (min:sec)	5:00
	Setting time	10:00
	Detail Reproduction (μm)	2
	Density (kg/m^3^)	1.06
	Tear resistance (N/mm^2^)	5
	Elastic recovery (%)	99.95%

**Table 2 polymers-14-04193-t002:** Material properties of the carbonyl iron particles.

Properties	Type 1	Type 2	Type 3	Type 4
Grade	CIP ER	CIP SQ-I	CIP HQ	CIP CS
Hard/soft	Hard	Soft	Hard	Soft
Particle size (μm)	4.5	3.8–5.4	2.0	6.0–7.0
Coating	SiO_2_	SiO_2_	-	-
Density (g/cm^3^)	~7.89	~7.89	~7.89	~7.89

**Table 3 polymers-14-04193-t003:** Composition and labels of the fabricated MREs.

S.N.	Matrix	CIP (Hard/Soft)	Coating	Volume%	Label
1	Silicone rubber	Hard	Yes	5%	H-C-5
2	Silicone rubber	Soft	Yes	5%	S-C-5
3	Silicone rubber	Hard	No	5%	H-NC-5
4	Silicone rubber	Soft	No	5%	S-NC-5
5	Silicone rubber	Hard	Yes	10%	H-C-10
6	Silicone rubber	Soft	Yes	10%	S-C-10
7	Silicone rubber	Hard	No	10%	H-NC-10
8	Silicone rubber	Soft	No	10%	S-NC-10
9	Silicone rubber	Hard	Yes	20%	H-C-20
10	Silicone rubber	Soft	Yes	20%	S-C-20
11	Silicone rubber	Hard	No	20%	H-NC-20
12	Silicone rubber	Soft	No	20%	S-NC-20
13	Silicone rubber	-	-	0%	Unfilled

The samples with hard CIPs are indicated with ‘H’, while the ‘S’ letter indicates soft CIPS. The coated samples are presented by ‘C’ and ‘N.C.’ for the noncoated samples. The value at the end of the labels represents the CIP volume fraction of the samples.

## Data Availability

The data presented in this study are available on reasonable request from the corresponding author.

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
