# Peer review of "Effect of Carbonyl Iron Particle Types on the Structure and Performance of Magnetorheological Elastomers: A Frequency and Strain Dependent Study"

_polymers, 2022, doi:10.3390/polym14194193_

Round 1

Reviewer 1 Report

Within this manuscript, various compositions of magnetorheological elastomers (MREs) with varying content of magnetic particles are exposed to experimental measurements to see if there is a relationship between storage and loss modulus with frequency and strain rate. Since the topic falls within the most important ones in this area and interesting data is reported in the manuscript, the reviewer believes this manuscript is capable enough to be given a second chance for reconsideration. Before the reviewer is able to choose a recommendation for present text, the authors are invited to answer bellow comments in detail:

1.       First off, the English writing is confusing and needs serious modifications.

2.       The abbreviations need to be introduced in their first appearance in the text. In some cases, this order is not payed attention. Check it through the manuscript.

3.       The Abstract is badly written. Of course, the readers will become totally confused while reading such a perplexing Abstract. This section seems like showcase of your entire article. Complete revision of this part is seriously required. Necessity, method, results, and benchmarks must be covered in a distinguishable order.

4.       The reviewer is curious to know whether couple stresses existing in the micro scale are regarded about the carbonyl iron particles.

5.       What was the reason for excluding the time effect on the dynamic mechanical analysis (DMA) results? My main concern is to address whether storage modulus and loss factor are functions of setting and working time intervals for instance.

6.       The reviewer could not find anything addressing the influence of the induced Lorentz force on the dynamical features of the MRE. What is the reason? What would be the effect based on your opinion?

Reviewer 2 Report

Comments on polymers-1886890

The manuscript entitled “Effect of Carbonyl Iron Particle Types on The Structure and Performance of Magnetorheological Elastomers: A Frequency and Strain Dependent Study” aims to examine the effect of the different magnetic particles on the performance of the MRE structures with the different CIPs concentrations and compositions of hard/soft and coated/noncoated magnetic particles. The loss and storage modulus of the samples are also verified under different excitation frequencies and strain amplitudes. The manuscript is well-written, results are well-discussed, however, there are several amendments required to be resolved before accepting it for publication which are disclosed below:

·         The authors should provide the full forms in the very starting, for example, CIPs in the Abstract.

·         The authors are also highly recommended to add the major results/findings of their work in the last paragraph of the Introduction.

·         In the Figure 4 caption, the authors wrote Figure 4 (a), what is Figure 4 (a), please recheck it.

·         Why do the authors stop the volume fraction of the fillers till 20 % ?? And what is the percolation of your samples ??

·         How do the authors assure the dispersion of particles in the matrix ?? The authors should provide some experimental results to assure the homogenous dispersion of particles in the matrix.

·         The authors are also suggested to show the microstructure of MRE samples.

·         Why do S-C and H-C samples have almost the same Relative MR effects??

·         The authors are highly suggested to show some SEM images of coated and uncoated particle samples for better understanding.

·         The authors are suggested to write a few sentences to propose some prospective applications of their work in the last paragraph before the Conclusions.

·         The authors are suggested to add some recently published references relevant to their work. Some references are important to understand the progress of silicone rubber-based composite materials and the advantages of silicone rubber: Composites Part A, 2022, 153, 106734. Please check all the reference formatting again.

·         The language expression in the text needs to be carefully checked and revised. There are a few grammatical mistakes.

Reviewer 3 Report

Recommendation: Major Revision

This paper has investigated the effect of carbonyl iron particle types on the structure and frequency and strain dependent performance of MRE. The main problem of this paper is the lack of in-depth discussion and related literature surveys. In addition, many important contributions in the related topic is completely missing in this manuscript. Probably the paper can be considered but after substantial revision with more discussions. The more specific comments are given here. 

1.     In introduction part, please include more recent literatures which reported the similar topics. There are many literatures which investigated the MR effect and tunable frequency range at the same time.

2.     For tunable MRE-based vibration absorber, the tunable and controllable frequency range is important. Please compare the vibration suppression range and MR effect of this result with the previous studies. The following paper will help the authors to discuss the MR effect and vibration suppression range.

(1) D.I. Jang et al, Designing an attachable and power-efficient all-in-one module of a tunable vibration absorber based on magnetorheological elastomer, Smart Mater. Struct. 27 (2018) 85009. https://doi.org/10.1088/1361-665X/aacdbd.

(2) K.H. Lee et al, Design of a stiffness variable flexible coupling using magnetorheological elastomer for torsional vibration reduction, J. Intell. Mater. Syst. Struct. 30 (2019) 2212–2221. https://doi.org/10.1177/1045389X19862378.

3.  During the curing process, the magnetization curing is used in this study? Because, it has been reported that the with and without magnetization curing can change the CIP-based chain structures, affecting the MR effect. Jang et al reported that the magnetization curing can change the CIP-formation, and they observed it using the electrical conductivity. Please review the following papers, and include them for more discussions.

(1) Fu, Y et al, Investigation into a Lightweight Polymeric Porous Sponge with High Magnetic Field and Strain Sensitivity. Nanomaterials 2022, 12, 2762. https://doi.org/10.3390/nano12162762

4.  There are many papers which used CIP to fabricate the MREs for designing tunable vibration absorbers. Although this paper observed the effect of CIP types on performance of MRE, the reviewers could not find any novelty and contribution of this paper compared to the previous studies. Please emphasize the novelty and contribution of this papers in some sections.      

Round 2

Reviewer 1 Report

The authors had done their best to modify the manuscript. The format-related issues are well modified now. Also, a much better Abstract is written. Although the authers' reply to the last three comments is not satisfactory, the article's improvement is clearly observed. Thus, this manuscript can be considered for publication based on the Editor's own preference.

Reviewer 3 Report

The authors have revised the manuscript considering the reviewer's comments. Thus, the present manuscript can be published in this journal.